# Evaluation of an Adaptive Soil Moisture Bias Correction Approach in the ECMWF Land Data Assimilation System

**David Fairbairn *** , **Patricia de Rosnay** and **Peter Weston**

European Centre for Medium-Range Weather Forecasts, Reading RG2 9AX, UK;
patricia.rosnay@ecmwf.int (P.d.R.)
* Correspondence: david.fairbairn@ecmwf.int

**Abstract:** Satellite-derived soil moisture (SM) observations are widely assimilated in global land data assimilation systems. These systems typically assume zero-mean errors in the land surface model and observations. In practice, systematic differences (biases) exist between the observed and modelled SM. Commonly, the observed SM biases are removed by rescaling techniques or via a machine learning approach. However, these methods do not account for non-stationary biases, which can result from issues with the satellite retrieval algorithms or changes in the land surface model. Therefore, we test a novel application of adaptive SM bias correction (BC) in the European Centre for Medium Range Weather Forecasts (ECMWF) land data assimilation system. A two-stage filter is formulated to dynamically correct biases from satellite-derived active ASCAT C-band and passive L-band SMOS surface SM observations. This complements the operational seasonal rescaling of the ASCAT observations and the SMOS neural network retrieval while allowing the assimilation to correct subseasonal-scale errors. Experiments are performed on the ECMWF stand-alone surface analysis, which is a simplified version of the integrated forecasting system. Over a 3 year test period, the adaptive BC reduces the seasonal-scale (observation−forecast) departures by up to 20% (30%) for the ASCAT (SMOS). The adaptive BC leads to (1) slight improvements in the SM analysis performance and (2) moderate but statistically significant reductions in the 1–5 day relative humidity forecast errors in the boundary layer of the Northern Hemisphere midlatitudes. Future work will test the adaptive SM BC in the full integrated forecasting system.

**Keywords:** soil moisture; data assimilation; bias correction

## 1. Introduction

Inherent biases exist when assimilating satellite-derived soil moisture (SM) observations in land surface models. Observation biases may result from instrument errors, vegetation effects and errors of representativeness, whilst model biases can originate from the model physics, parameterisations, initial conditions and atmospheric forcing [1,2]. Biases can be addressed by data assimilation (DA) when it is not possible to correct the source directly.

Commonly in atmospheric and ocean DA, a variational bias correction (BC) approach is employed for satellite radiances, whereby the corrections for the computed forward radiances are updated within the cost function minimisation [3,4]. Additionally, a network of anchor observations is employed as a reference (e.g., radiosondes) to prevent the analysis from drifting to the model bias [5]. Land surface models are much more heterogeneous than atmospheric or ocean models, and point-wise reference SM measurements are not generally representative of the large-scale footprint associated with model- or satellite-derived products [6]. Without an accurate ground reference, numerical weather prediction (NWP) centres typically assume that the land surface model is perfect and all the SM biases belong to the observations.

Traditionally, in NWP, rescaling approaches have been used to map the SM observations to the model climatology. Commonly, a point-wise cumulative distribution function (CDF) matching approach is employed to rescale the mean and variance of the observations to the model equivalent [7–11]. Thus, the model climatology is preserved, and the assimilation is designed to correct temporal anomalies. At the UK Met Office, a similar rescaling approach is adopted, where a fraction of the Advanced Scatterometer (ASCAT) soil wetness anomaly (from its observational climate) is added to the model's monthly mean surface SM [12]. The model climatology is then changed by the observation, and it is this value that is assimilated.

Over the last decade, there has been some interest in using more information from SM observations, including the mean and variance. Kumar et al. [13] applied a parameter estimation technique to calibrate a land surface model to the climatology of synthetic SM observations. They compared this approach with the traditional CDF matching rescaling approach described above. They found that both techniques effectively removed model observation biases and delivered similar levels of skill in the surface and root zone SM analyses. Machine learning methods have also been advocated in recent years, including neural networks that convert level one satellite data to level two SM [14,15]. Aires et al. [15] compared a global inverse neural network (NN) technique with the traditional point-wise CDF matching approach for ASCAT SM BC. Both approaches were trained and calibrated using the same land surface model. They found that the two approaches worked better in different regions of the world, and merging the two methods delivered the most effective global BC.

At the European Centre for Medium Range Weather Forecasts (ECMWF), ASCAT- and soil moisture ocean salinity (SMOS)-derived SM observations are both assimilated operationally in the SM analysis using a simplified extended Kalman filter (SEKF) [16,17]. The ASCAT SM observations are bias-corrected via the aforementioned CDF matching approach. On the other hand, the SMOS SM is derived from the level one product using an NN trained on the ECMWF operational analysis [14]. Whilst these methods work well in many practical applications, they require large calibration and training datasets, and the assumption of stationary biases is often suboptimal. Biases in the observations can change over time due to issues with the instruments and modifications in the retrieval algorithms, whilst land surface models are also updated periodically. Both of these factors change the observation-model climatology and necessitate a recalibration of the climatological BC. These recalibrations require a long enough sampling period to capture the climatology (typically more than 3 years), which is not always available.

In this study, a novel adaptive BC application is tested for ASCAT and SMOS assimilation. It is designed to capture non-stationary biases and is complementary to the operational ASCAT CDF matching and SMOS NN. The two-stage filter was introduced by Dee and Silva [18], where the filter and the adaptive BC updates are performed independently. Applications can be found for the atmosphere [1], ocean [19,20] and land [21–24]. Table 1 summarises the observation and model types for the referenced land surface applications and this study. The filter was originally formulated to bias-correct the model according to the assumption that the observations were unbiased. Draper et al. [25] reformulated the two-stage filter to bias-correct the observations according to the assumption that the model was unbiased. They successfully applied the BC to skin-temperature assimilation in the catchment and surface modelling component of the Goddard Earth Observing System. A similar approach is adopted here in the context of satellite-derived SM assimilation in the ECMWF integrated forecasting system (IFS). As far as the authors are aware, this study is the first application of the two-stage filter for satellite-derived SM assimilation in an NWP system. The aim of this study is to evaluate the impact of the adaptive BC over a 3 year period, focusing on (1) the observation—forecast $(O - F)$ departures, (2) the SM analysis performance against in situ data and (3) the NWP forecast skill for low-level atmospheric temperature and humidity.

**Table 1.** Example land surface applications of the two-stage adaptive BC approach. The abbreviations "land-atmos", "obs" and "temp" are short for "coupled land-atmosphere model", "observations" and "temperature", respectively. The last entry refers to the present study.

| Study | Observation Type | Model Type | Region | BC Variable |
|---|---|---|---|---|
| Bosilovich et al. [21] | Skin temp | Land-atmos | Global | Model surface temp |
| De Lannoy et al. [22] | In situ SM | Land only | Regional | Model SM |
| Reichle et al. [23] | Skin temp | Land only | Global | Model surface temp |
| Pauwels et al. [24] | Streamflow | Hydrological | Regional | Streamflow model and obs |
| Draper et al. [25] | Skin temp | Land only | Regional | Skin temp model and obs |
| Present | SM | Land-atmos | Global | SM obs |

## 2. Materials and Methods

### 2.1. The Bias-Free SEKF Soil Moisture Analysis

Following Boussetta et al. [26], the land surface component of the ECMWF IFS is now referred to as ECland. This includes the land surface model and DA. The operational SM analysis of ECland is based on a point-wise SEKF with 12 h assimilation windows implemented in 2010 to assimilate proxy 2 m height (screen-level) observations of the temperature (T2m) and relative humidity (RH2m) [16]. These proxy observations come from the analysed T2m and RH2m states at the synoptic times (every 6 h), which are derived by assimilating SYNOP screen-level observations with a 2D optimal interpolation (OI) scheme. ASCAT-derived SM observations have been assimilated in operations since 2015, followed by SMOS NN assimilation since 2019 [14]. Up to 8 observations can be assimilated for each grid point per assimilation window, with a maximum of 2 each for ASCAT, SMOS, T2m and RH2m. The top 3 layers (top metre) of the ECland model are analysed with depths of 0–7 cm, 7–28 cm and 28–100 cm from top to bottom. Using the notation of Ide et al. [27], the point-wise SM analysis state update at time ($t_i$) is expressed as

$$\mathbf{x}^a(t_i) = \mathbf{x}^b(t_i) + \mathbf{K}_i[\mathbf{y}^o(t_i) - H_i(\mathbf{x}^b)], \quad (1)$$

where the superscripts $a$, $b$, $o$ and $i$ denote the analysis (of dimension $n$), background (of dimension $n$), observations (of dimension $p$) and time step, respectively. The observation operator $H$ maps the model state to the observation space. As the preprocessed ASCAT and SMOS NN observations are already in the modelled volumetric SM units, $H_i(\mathbf{x}^b)$ is simply the model value in the top layer at the nearest grid point and time step to the observation. This simplification is reasonable considering the top layer (7 cm depth) is shallow enough to compare with the observation penetration depth (1–3 cm). The weights of the observations and background state in the analysis update are determined by the Kalman gain matrix $K$ (of dimension $n \times p$):

$$\mathbf{K}_i = [\mathbf{B}^{-1} + \mathbf{H}_i^T\mathbf{R}^{-1}\mathbf{H}_i]^{-1}\mathbf{H}_i^T\mathbf{R}^{-1}, \quad (2)$$

where $\mathbf{R}$ (of dimension $p \times p$) and $\mathbf{B}$ (of dimension $n \times n$) represent the observation and background error covariance matrices respectively, which are assumed to be static and diagonal (uncorrelated). The SM background errors are prescribed with values $\sigma_b = 0.01$ m$^3$/m$^3$, where $\sigma$ is the standard deviation. The ASCAT- and SMOS-derived SM observations are prescribed values of $\sigma_{ascat} = 0.05$ m$^3$/m$^3$ and $\sigma_{smos} = 0.02 + \epsilon$, respectively, where $\epsilon$ is a situation-dependent uncertainty output provided by the NN itself. The screen-level observations are assigned values of $\sigma_{T2m} = 1$K and $\sigma_{RH2m} = 4$% [28]. Flow-dependent uncertainty information from an ensemble of data assimilations (EDA, Bonavita et al. [29]) is implicitly propagated from the observations to the analysis state via the Jacobian matrix $\mathbf{H}$ (of dimension $p \times n$). The Jacobian linking the $k$th observation to the modelled SM layer $j$ is given by

$$\mathbf{H}_{k,j} = \frac{cov(H_k(\mathbf{x}^{eda}), \mathbf{x}_j^{eda})}{var(\mathbf{x}_j^{eda})}.c_j, \quad (3)$$

where "*cov*" stands for "covariance", "*var*" stands for "variance" and "*eda*" implies that the EDA ensemble is used. Tapering coefficients $c_j = 1/(1 + (j-1).\alpha_{sekf})$ were empirically derived to optimise the impact for the different SM layers ($j = 1, 2, 3$), with $\alpha_{sekf} = 0.6$. In the following assimilation window, the background (prior) state is a model simulation initialised by the current analysis state:

$$\mathbf{x}^b(t_{i+1}) = M[\mathbf{x}^a(t_i)], \tag{4}$$

where $M$ is the nonlinear coupled forecast model.

### 2.2. The Two-Stage Bias Filter

In the bias-aware SM analysis, the biased observations ($\tilde{\mathbf{y}}^o$) are partitioned into the analysed bias ($\mathbf{z}^a(t_i)$) and the non-biased term ($\mathbf{y}^o(t_i)$):

$$\tilde{\mathbf{y}}^o(t_i) = \mathbf{z}^a(t_i) + \mathbf{y}^o(t_i). \tag{5}$$

The state update then includes the bias-corrected observations:

$$\mathbf{x}^a(t_i) = \mathbf{x}^b(t_i) + \mathbf{K}_i[\tilde{\mathbf{y}}^o(t_i) - \mathbf{z}^a(t_i) - H_i(\mathbf{x}^b)]. \tag{6}$$

In this study, we only investigate biases in the ASCAT and SMOS NN observations, and therefore we assume that the proxy screen-level observations are bias-free. For simplicity, the ASCAT and SMOS NN bias updates are performed separately over each gridpoint (i.e., they are assumed to be uncorrelated). The bias update is calculated following the approach of Draper et al. [25] as follows:

$$\mathbf{z}_l^a(t_i) = \mathbf{z}_l^b(t_i) + \mathbf{L}_{i,l}[\tilde{\mathbf{y}}_l^o(t_i) - \mathbf{z}_l^b - H_{i,l}(\mathbf{x}^b)], \tag{7}$$

$$\mathbf{L}_{i,l} = [\mathbf{B}_l^{\mathbf{z}}][\mathbf{R}_l + \mathbf{B}_l^{\mathbf{z}} + \mathbf{H}_{i,l}\mathbf{B}\mathbf{H}_{i,l}^T]^{-1}, \tag{8}$$

where the subscript $l$ indicates the observation type and $\mathbf{B}_l^{\mathbf{z}}$ is the prior observation bias covariance matrix. The matrix $\mathbf{R}_l$ represents the random part of the observation error covariance matrix using the bias-free values of $\mathbf{R}$ defined for the ASCAT and SMOS NN in Equation (2). The Jacobian matrix $\mathbf{H}_{i,l}$ is the subset of the Jacobian matrix $\mathbf{H}_i$. Following [18,22], $\mathbf{B}_l^{\mathbf{z}}$ is a diagonal matrix which is proportional to the SM background error covariance matrix:

$$\mathbf{B}_l^{\mathbf{z}} = \frac{\gamma}{1-\gamma}\mathbf{H}_{i,l}\mathbf{B}\mathbf{H}_{i,l}^T, \tag{9}$$

where $\gamma$ is a scalar parameter. After putting Equation (9) into Equation (8) and rearranging it, the Kalman gain can be expressed as

$$\mathbf{L}_{i,l} = \gamma\mathbf{H}_{i,l}\mathbf{B}\mathbf{H}_{i,l}^T[\mathbf{H}_{i,l}\mathbf{B}\mathbf{H}_{i,l}^T + (1-\gamma)\mathbf{R}_l]^{-1}. \tag{10}$$

Increasing the value of $\gamma$ to be between 0 and 1 effectively reduces the memory of the bias. In this study, the value of $\gamma = 0.25$ was chosen empirically in order to have a long enough memory to capture seasonal-scale changes in the bias. After concatenating the updated bias correction for each observation type from Equation (7) into a single vector $\mathbf{z}^a(t_i)$ (of dimension $p$), the SM analysis is calculated with Equation (6). Recall that the screen-level observations are assumed to be bias-free, and therefore the elements of $\mathbf{z}^a(t_i)$ are set to zero for these observations. Over grid points where no observations are available per type $l$, the bias state is not updated. If only one observation is available, then Equation (7) is a scalar update. In cases where 2 observations of the same type are available for assimilation, $\mathbf{z}_l^a(t_i)$ is a vector with a size of 2. In the SM analysis, the observations are assumed to be

representative of the model grid point, and hence a weighted average of the bias update is stored and used for the following cycle:

$$\tilde{z}_l^a(t_i) = \frac{1}{K} \sum_{k=1}^{K} z_{l,k}^a(t_i), \qquad (11)$$

where $k$ is the observation index and $K$ is the total number of observations of type $l$ in the grid point (maximum of 2 for the ASCAT or SMOS NN). In practice, overlapping observations are mainly found in high latitudes, and the averaging of the bias update has little impact on the SM performance. The persistence model is employed for the bias forecast, which is a reasonable assumption if the bias evolves slowly between cycles [18,22]. Hence, the background bias state for the following cycle is defined as

$$\tilde{z}_l^b(t_{i+1}) = \tilde{z}_l^a(t_i). \qquad (12)$$

The background bias states for each grid point are then used for all the observations of type $l$ in the gridpoint for the following cycle:

$$\mathbf{z}_l^b(t_{i+1}) = (\mathbf{1}_K)^T \tilde{z}_l^b(t_{i+1}), \qquad (13)$$

where $(\mathbf{1}_K)^T$ is a vector of ones of dimension $K$.

### 2.3. Satellite-Derived Soil Moisture Observations and Preprocessing

An introduction to the ASCAT- and SMOS-derived SM data is given below along with the current SM bias correction techniques.

Active C-band scatterometer data are provided by the ASCAT sensors on board the Metop satellites, with a global revisit time of 1–3 days. These observations are converted into a liquid soil wetness percentage using the change detection approach [30,31]. The change detection approach assigns dry and saturated soils to the historically lowest and highest backscatter coefficient values, respectively. This approach relies on a long time series for each pixel. The EUMETSAT Satellite Applications Facility for Hydrology (H SAF) has employed this technique for the development and production of level 2 ASCAT-derived surface SM products. At the ECMWF, the 25 km resolution product is currently assimilated into the operational NWP system [32]. During the study period (2019–2022), the ASCAT from Metop-B was available throughout. ASCAT Metop-C assimilation was introduced in October 2019, and Metop-A assimilation was retired in November 2021. ASCAT SM is measured as a soil wetness percentage (between 0 and 100%), and the modelled SM is expressed as the volumetric soil water content. Cumulative distribution function (CDF)-matching bias correction rescales the ASCAT level 2 SM such that the observed CDF matches the modelled SM CDF. Systematic differences between the observations and the model are subsequently removed, and the observations are effectively converted into volumetric units. At the ECMWF and various other centres, only the first two moments of the observation CDF, the mean and variance, are rescaled to match the model mean and variance [9]. A CDF matching which accounts for seasonal variability was implemented by [33,34] and has been adopted by the ECMWF in the simplified extended Kalman filter (SEKF) SM analysis [16] as well as for SMOS brightness temperature bias correction [11]. Hereafter, this method will be called "seasonal rescaling". The seasonal rescaling parameters are derived for each calendar month using a 3 month moving average. For example, for the month of May, the parameters are derived based on the data from April, May and June over a multi-year calibration period. The operational seasonal rescaling parameters for IFS cycle 47r3 are based on rescaling the ASCAT-A/B-derived SM to the ERA5 surface soil moisture (SSM) over the period of 2009–2018 [35]. The ASCAT observations undergo a quality control check prior to assimilation, which screens observations during frozen conditions, over mountainous regions and where the estimated noise exceeds 15%.

The SMOS SM observations are derived from the L-band brightness temperature (Tb) using a neural network approach, which is summarised below, but full details can be found in Rodríguez-Fernández et al. [14,36]. They are trained and validated using the high-resolution ECMWF operational SM analysis over the period of 2015–2020. Firstly, the SMOS Tb observations are colocated with the model grid points in space and time. In order to maximise the data availability, Tb observations are extracted from incidence angles ranging from 30 to 45°. Twelve predictors are used, consisting of 6 Tb and 6 SSM linear expectations and 3 angular bins. Furthermore, the ECMWF operational snow depth and temperature analysis are used to filter out data during frozen conditions. After filtering, 60% of the samples are used for training, 20% are used for an evaluation of the training dataset, and a further 20% are used for the validation of the SMOS NN-derived SM. A gradient backpropagation approach is employed for the training using the Levenberg–Marquardt algorithm [37]. The architecture consists of a 2 layer NN with 1 hidden layer and 5 neurons. The NN is trained globally, which means that while the global biases between the SMOS SM observations and the model SM are small, significant regional biases remain.

### 2.4. ECland Surface Model

The ECLand surface model is used in the IFS [26]. It is inherited from the Hydrology Tiled ECMWF Scheme for Surface Exchanges over Land (HTESSEL; [38]). ECland models land surface processes including SM, snow and vegetation. Each grid point is divided into 8 tiles representing different land cover types, including vegetation types, soil and snow cover, using data from the US Department of Agriculture Global Land Cover Climatology (GLCC) map [39]. The seasonal vegetation cycle is constructed from the monthly leaf area index climatology [40]. Soil moisture is represented by 4 vertical layers with thicknesses of 7 cm, 21 cm, 72 cm and 189 cm from top to bottom. The vertical SM exchanges are based on the equation from Richards [41]. The HTESSEL model improved on the original TESSEL model from Viterbo and Beljaars [42] by introducing spatial variability into the soil texture. In addition, fast surface runoff is represented by a variable infiltration capacity. A multi-layer snow scheme controls the snowpack evolution [43]. The HTESSEL model will be improved in a future version of ECLand, which will include higher vertical discretisation and a much thinner surface layer [26].

### 2.5. The Stand-Alone Surface Analysis

This study makes use of the stand-alone surface analysis (SSA) of Fairbairn et al. [35]. A flow diagram of SSA is given in Figure 1. In each cycle, the atmospheric initial conditions for the coupled model are forced by an external atmospheric analysis, which avoids the computational burden of running the 4D-Var atmospheric analysis. In this study, atmospheric forcing consists of the the ERA5 reanalysis fields [44]. However, land–atmosphere feedback is provided by the coupled forecast model between cycles.

Other than the atmospheric analysis, SSA benefits from most of the functionalities in the IFS. The land surface analysis, as described by ECMWF [28], includes a 2D OI for T2m, RH2m and the snow depth analysis. Additionally, the soil temperature (ST) is analysed with a 1D OI. As explained by Haseler [45], two 12 h assimilation windows run from 9:00 p.m. to 9:00 a.m. and 9:00 a.m. to 9:00 p.m. UTC, which provide the initial conditions for the 10 day coupled NWP forecasts at 12:00 a.m. UTC and 12:00 p.m. UTC, respectively.

Fairbairn et al. [35] compared the SSA system with the operational weakly coupled land–atmosphere DA system. Firstly, they analysed the relative impacts of an important land DA change on the NWP forecasts of the two systems. Secondly, they compared the computational efficiency of the two systems. Compared with weakly coupled DA, SSA demonstrated (1) a reduced but largely consistent impact of the land DA change on NWP forecasts and (2) a substantial (>70%) reduction in the computing time. SSA is particularly useful for running multi-year experiments in a reasonable time frame, which is important for monitoring the long timescales associated with SM bias correction.

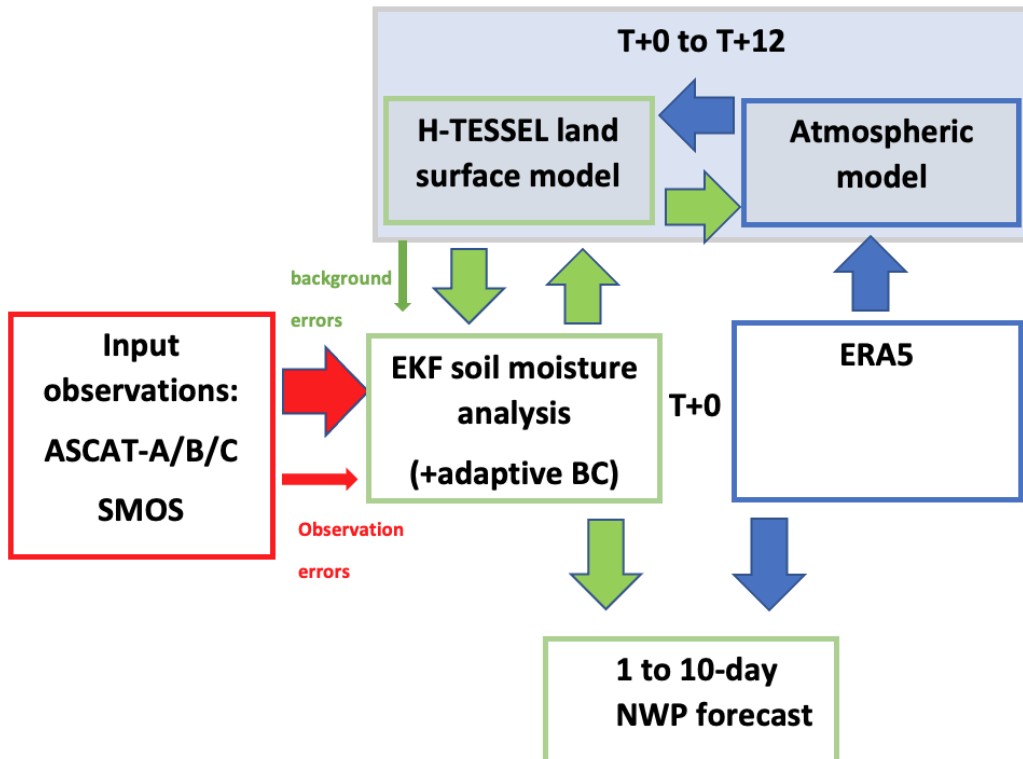

**Figure 1.** Flow diagram of the stand-alone surface analysis (SSA) used in the experiments.

### 2.6. Experiments

Five SSA experiments were validated globally over a 3 year period (1 January 2020–31 December 2022) after a 1 yr spin-up (1 January 2019–31 December 2019). All the experiments assimilated both the ASCAT- and SMOS-derived SM observations. Table 2 summarises the differences between the experiments: (1) C is the control with the operational configuration of ASCAT and SMOS assimilation without adaptive BC, (2) $E_A$ includes ASCAT adaptive BC, (3) $E_S$ includes SMOS adaptive BC, and (4) $E_{A,S}$ includes both ASCAT and SMOS adaptive BC. Experiments (2) and (3) were individual assessments of ASCAT and SMOS adaptive BC, respectively. On the other hand, experiment (4) assessed the combination of ASCAT and SMOS adaptive BC. If the ASCAT and SMOS adaptive BC had a complementary impact, then experiment (4) should have had a smaller model observation bias than (1) or (2). In experiments (1)–(4), ASCAT seasonal rescaling was applied before the adaptive BC. Experiment (5) ($E_{A,S}{}^*$) was designed to test whether the adaptive BC alone could improve on the seasonal rescaling for the ASCAT. In experiment (5), the level 2 ASCAT SM for each grid point was converted to volumetric units by rescaling the percentage to a soil wetness index (SWI) between 0 and 1 and then multiplying the SWI by the modelled saturation value. Each experiment was initialised by ERA5 reanalysis at 12:00 a.m. UTC on 1 January 2019. In subsequent cycles, the atmospheric analysis was initialised by ERA5 (as described in Section 2.5). The land surface analysis and the coupled model were performed with cycle 47r3 of the ECMWF IFS. The experiments were run on a cubic octahedral reduced Gaussian grid at Tco319 (approximately 31 km resolution). This resolution is quite close to the ERA5 native resolution, avoiding issues with spatial interpolation of the atmospheric fields.

**Table 2.** List of experiments used to test the adaptive SM bias correction. The experiment $E_{A,S}$* is equivalent to $E_{A,S}$ except that ASCAT seasonal rescaling is not applied.

| Experiment | C | $E_A$ | $E_S$ | $E_{A,S}$ | $E_{A,S}$* |
|---|---|---|---|---|---|
| ASCAT adaptive BC | False | True | False | True | True |
| SMOS adaptive BC | False | False | True | True | True |
| ASCAT seasonal rescaling | True | True | True | True | False |

*2.7. SM and ST Validation Approach*

The surface (0–7 cm) and root zone (0–100 cm) layers of SM and ST were validated using sparse in situ data from the International Soil Moisture Network (ISMN [46]). Observations were extracted at the nearest hour to the analysis times (12:00 a.m. and 12:00 p.m. UTC $\pm$30 min) from 7 networks over the United States, Europe and Australia. The North American networks consisted of the U.S. Climate Reference Network (USCRN [47]) and the U.S. Department of Agriculture's Soil Climate Analysis Network (SCAN) and Snowpack Telemetry (SNOTEL) networks [48]. In Europe, they consisted of the Soil Moisture Observing System–Meteorological Automatic Network Integrated Application (SMOSMANIA, [49]) in southwestern France, the Soil Moisture Measurement Stations Network of the University of Salamanca (REMEDHUS) in the central sector of the River Duero basin in Spain [50] and the Network of Terrestrial Environmental Observatories in Germany (TERENO, [51]). In Australia, the Murrumbidgee Soil Moisture Monitoring Network data set was used (OZNET, [52]). The depths, locations and number of stations in each network are listed in Table 3. Although the validation of model grid point data using in situ observations is affected by representativeness and instrument errors, the locations of these networks represent a wide range of vegetation and soil types. Similar validations have been performed at the ECMWF [35,53] and elsewhere [54,55] for validating global SM analyses and satellite products. The in situ SM observations were compared with the root zone soil moisture (RZSM) over the top metre of soil by taking a vertical average, with weights that were proportional to the spacings between the sensor depths (as in [55]). For the SNOTEL network, SMOSMANIA, OZNET and TERENO, the deepest measurement was assumed to represent the depth of the observation down to 1 m. In each network, the top measurement at a 5 cm depth was compared to the surface layer of the SM analysis (0–7 cm depth). The observations underwent a rigorous quality control check by the ISMN facility [56], with flags during frozen conditions and implausible SM values (e.g., spikes). Additional quality control screening was implemented in this study, where the analysed ST was below 4 °C, which reduced the risk of frozen conditions. Furthermore, a minimum of 50 observations was required for each station during the validation period to reduce sampling errors.

**Table 3.** Reference data sets used for the validation in this study. ECMWF* refers to the ECMWF operational deterministic analysis interpolated from 9 km to 31 km resolution.

| Name | Reference Type | Vertical Depths or Levels | Spatial Res. | Num. of Stations |
|---|---|---|---|---|
| SMOSMANIA | In situ SM or ST | 5, 10, 20 and 30 cm depth | Point-wise | 20 stations, France |
| SCAN | In situ SM or ST | 5, 10, 20, 50 and 100 cm depth | Point-wise | 133 stations, US |
| USCRN | In situ SM or ST | 5, 10, 20, 50 and 100 cm depth | Point-wise | 106 stations, US |
| SNOTEL | In situ SM or ST | 5, 10, 20 and 50 cm depth | Point-wise | 292 stations, US |
| REMEDHUS | In situ SM or ST | 5 cm depth | Point-wise | 15 stations, Spain |
| OZNET | In situ SM or ST | 4, 15, 45 and 75 cm depth | Point-wise | 13 stations, Australia |
| TERENO | In situ SM or ST | 5, 20 and 50 cm depth | Point-wise | 1 station, Germany |
| ECMWF* | Air temp analysis | 137 levels (1–1000 hPa) | 31 km | Global analysis |
| ECMWF* | Air RH analysis | 137 levels (1–1000 hPa) | 31 km | Global analysis |

The validation metrics consisted of the Pearson R anomaly correlation coefficient ($R_{ano}$), unbiased root mean square error (UbRMSE) and the bias, as described by Fairbairn et al. [35]. These metrics offer complementary validation of the SM perfor-

mance [57]. It is acknowledged that errors of representativeness can cause large differences in the absolute values between the SM analysis and the in situ data for some stations. However, the Pearson R correlation coefficient is often used for SM validation as it is sensitive to the temporal variability rather than the absolute SM values. However, the Pearson R correlation coefficient is affected by the seasonal SM cycle. For $R_{ano}$, these seasonal-scale correlations were removed by subtracting the centred 35 day moving average from the time series. As in the work of Albergel et al. [53], the $p$ value provided a measure of the significance of the correlations, and the scores for each station were only retained where the $p$ value was less than 0.05. Furthermore, a Fisher z transform with a lag 1 autocorrelation was implemented to find the 95% confidence interval (CI) while mitigating the autocorrelations associated with the seasonal cycle [54].

### 2.8. Atmospheric Validation Approach

In line with the ECMWF NWP set-up, 10 day coupled forecasts were initialised at 12:00 a.m. and 12:00 p.m. UTC for all the experiments. The atmospheric forecasts were validated against the ECMWF operational analysis. Following Geer [58], the normalised RMSE differences (dRMSE) were used to validate the relative performance of the experiment forecasts compared to the control forecasts:

$$\text{dRMSE} = \frac{\left\| \mathbf{z}^e_{t:t+T} - \mathbf{z}^r_{t+T} \right\| - \left\| \mathbf{z}^c_{t:t+T} - \mathbf{z}^r_{t+T} \right\|}{\left\| \mathbf{z}^c_{t:t+T} - \mathbf{z}^r_{t+T} \right\|}, \tag{14}$$

where $\mathbf{z}^c_{t:t+T}$ ($\mathbf{z}^e_{t:t+T}$) represents the control (experiment) forecasts of a length $T$ from analysis time $t$ and $\mathbf{z}^r_{t+T}$ is the reference at time $t + T$, while $\|.\|$ in this case is the RMSE of the time series, which is calculated independently for each grid point. The reference was provided by the ECMWF high-resolution deterministic analysis. Furthermore, the difference in mean error between the experiments and the control was computed:

$$\text{dME} = \overline{\mathbf{z}^e_{t:t+T} - \mathbf{z}^r_{t+T}} - \overline{\mathbf{z}^c_{t:t+T} - \mathbf{z}^r_{t+T}}. \tag{15}$$

## 3. Results

### 3.1. Internal DA Diagnostics

Firstly, the impact of the adaptive BC on the ASCAT SM $(O - F)$ departures was evaluated. Recall that the departures were calculated as $(\mathbf{y}^o - H(\mathbf{x}^b))$ in the control and $(\tilde{\mathbf{y}}^o - \mathbf{z}^a - H(\mathbf{x}^b))$ in the adaptive BC experiments. Figure 2a shows the global mean monthly ASCAT SM departures over 2019–2021 for the control (C) and the experiment $E_{A,S}$. Whilst the mean departure oscillated close to zero in the first few months, it then trended upwards in C. At the same time, the BC $(-\mathbf{z}^a)$ applied to $E_{A,S}$ trended downwards, which resulted in less biased departures. The spin-up for the BC took about a year, after which the absolute departures (Figure 2b) were about 10–20% larger in C compared with $E_{A,S}$, thus confirming that the bias was a substantial part of the departure magnitude. The spin-up period was dependent on the seasonal variability in the SM departures and the memory of the bias. Recall that the bias memory is dependent on the parameter $\gamma$ in Equation (9). Figure 2c displays the used ASCAT SM observation count. Up to 6% more observations were assimilated in $E_{A,S}$ compared with C, with the largest difference observed during summer and autumn in the Northern Hemisphere, a period of active ASCAT assimilation in this region. Further investigation revealed that the quality control was rejecting a greater number of observations in C compared with $E_{A,S}$ when the departure size exceeded the maximum threshold of 0.1 $\text{m}^3/\text{m}^3$. Note that the BC update was still applied when the departure size exceeded this threshold, which enabled systematic large departures to be corrected. Figure 2c also demonstrates a substantial increase in the peak observation count after the introduction of ASCAT-C and a similarly large reduction in the peak count following the retirement of ASCAT-A, which highlights the non-stationary nature of satellite observations.

Figure 2d–f demonstrates the impact of the adaptive BC on the SMOS SM ($O - F$) departures. In Figure 2d, the mean global bias in C is highly variable, ranging from slightly negative to highly positive. Experiment $E_{A,S}$ responded with a negative BC, although it sometimes overcorrected in the negative direction. Nevertheless, the absolute departures were about 20–30% smaller in $E_{A,S}$ than C (Figure 2e), which indicates that the BC was mainly effective. Overall, up to 12% more SMOS SM observations were assimilated when adaptive BC was turned on (Figure 2f), with the biggest gap again seen in the summer in the Northern Hemisphere.

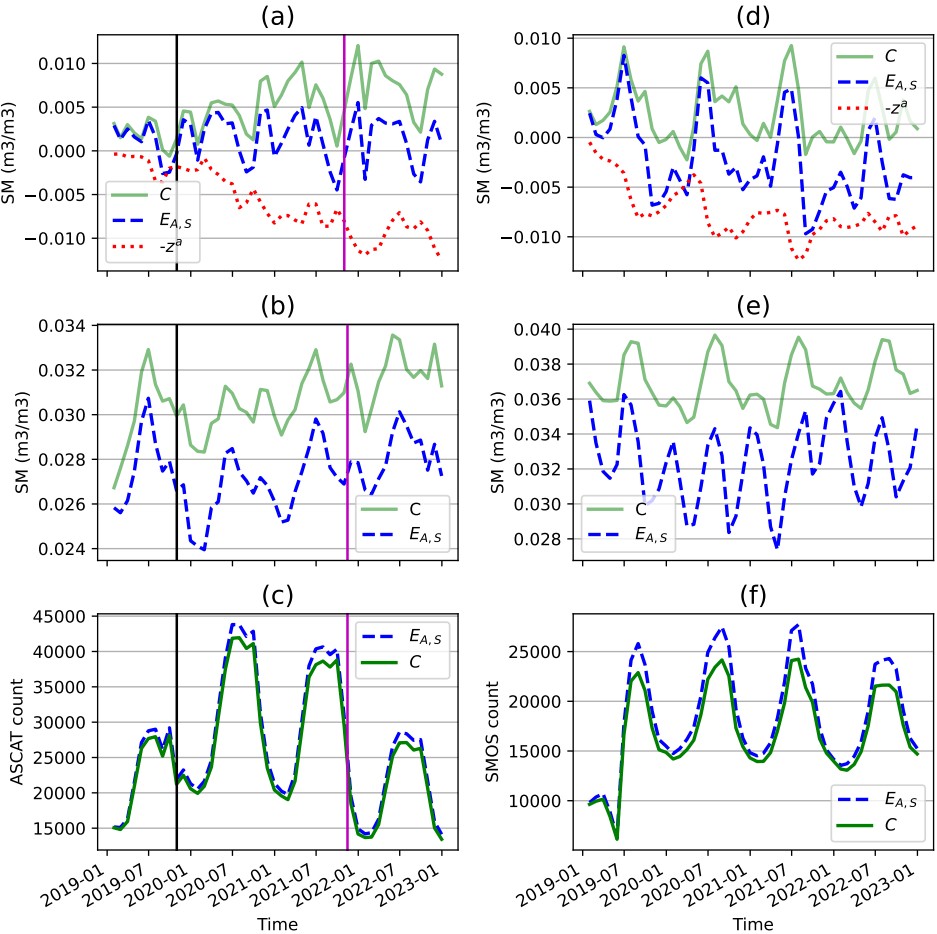

**Figure 2.** (**a**) Monthly and globally averaged ASCAT SM departures (m$^3$/m$^3$) for the control (C) and experiment ($E_{A,S}$) during 2019–2022. The start date of ASCAT-C assimilation is indicated by the first vertical line, and the retirement date of ASCAT-A assimilation is given by the second vertical line. Also shown is the BC term ($-z^a$) estimated by $E_{A,S}$. (**b**) Monthly and globally averaged absolute ASCAT SM departures (m$^3$/m$^3$). (**c**) Monthly and globally averaged count of the ASCAT observations assimilated in the control and the experiment. Plots (**d**–**f**) are SMOS SM equivalents to (**a**–**c**), respectively.

Next, the average ASCAT and SMOS SM departures were evaluated over August 2022 in Figure 3. In the control, the ASCAT had mainly positive departures over eastern Europe. On the other hand, the SMOS SM departures were generally positive over South America and South Africa. Large departures could be seen over some areas that are known to have quality control issues, such as the Andes mountain range in South America for SMOS SM and across the high latitudes for both the SMOS and ASCAT. Evidently, the BC reduced the departure magnitudes in these areas. Upon closer inspection, the BC was more active over some regions than others. For example, the strong negative SMOS SM departures over India and the Sahel region of Africa were barely corrected by the adaptive BC. Further

analysis has shown that the negative departure patterns in these regions only lasted for a few months, which was not long enough for the adaptive BC to spin up.

In Figure 4, the mean SM increments are shown for the surface layer (0–7 cm depth) over August 2022. The patterns were generally similar for C and $E_{A,S}$. However, in a latitude band from about 55° to 80° north, there were smaller magnitude increments in $E_{A,S}$ compared with C. In particular, northeast Asia showed a reduced positive signal when the adaptive BC was active. However, the impact of the adaptive BC on the increments in the lower latitudes (<55° north) was generally small, even in regions where the BC was large (e.g., South America). For the RZSM (top metre of soil), the depth-integrated increments were also impacted in the higher latitudes by the BC but to a lesser extent than the surface layer.

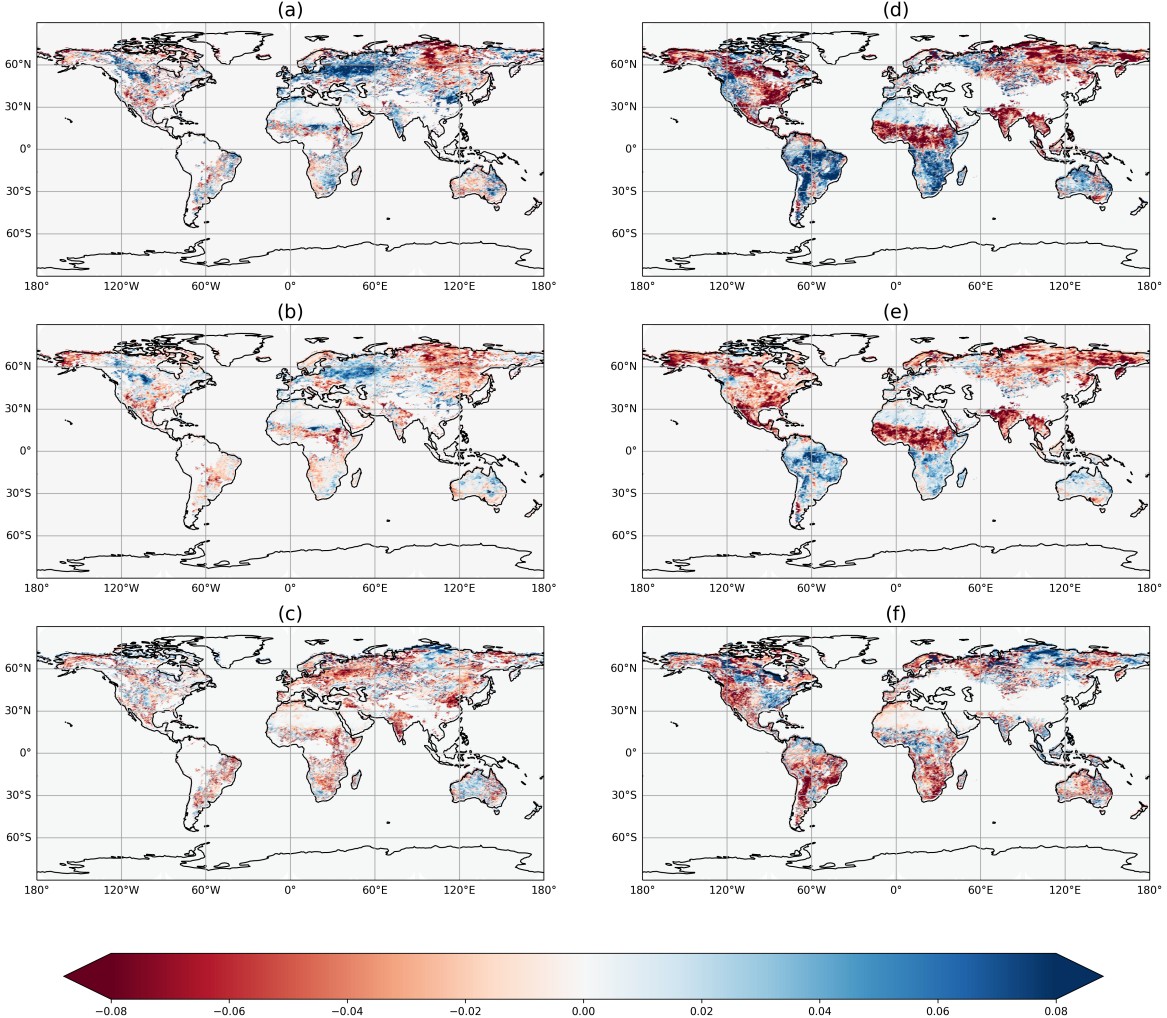

**Figure 3.** Mean ASCAT SM departures for (**a**) C and (**b**) $E_{A,S}$. The ASCAT SM bias predicted by $E_{A,S}$ is shown in (**c**). Plots (**d–f**) are SMOS SM equivalents to plots (**a–c**), respectively. All plots were averaged over August 2022, and units are in ($m^3/m^3$).

Focusing on a point in central eastern Australia with latitude 25° S and longitude 140° E, Figure 5 shows the time series of the SMOS SM ($O - F$) departures for (a) C and (c) $E_{A,S}$. Over the 4 year period, the departures were almost entirely positive for C, which demonstrates a local positive bias in the SMOS NN observations with respect to the model. The time series was effectively shifted in the negative direction by the BC while preserving the temporal variability in the departures. Figure 5b,d presents the distribution of the innovations normalised by ($\mathbf{HBH}^T + \mathbf{R}$) for C and $E_{A,S}$ respectively. These normalised innovations were computed according to Equation (1) of Desroziers et al. [59]. The positive

bias in C was addressed by a negative shift in the distribution for $E_{A,S}$. Neither distribution followed a Gaussian curve, which might be related to the nonlinear nature of the land surface model and the associated background and observation uncertainties.

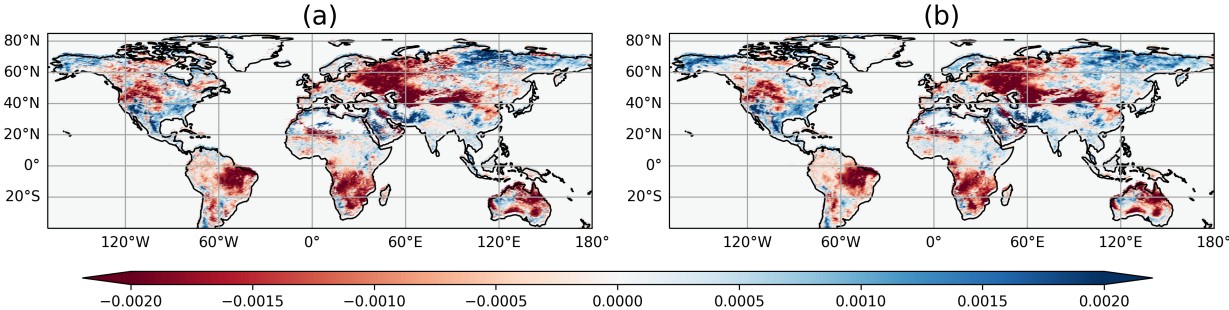

**Figure 4.** Mean SSM increments ($m^3/m^3$) over August 2022 for (**a**) C and (**b**) $E_{A,S}$.

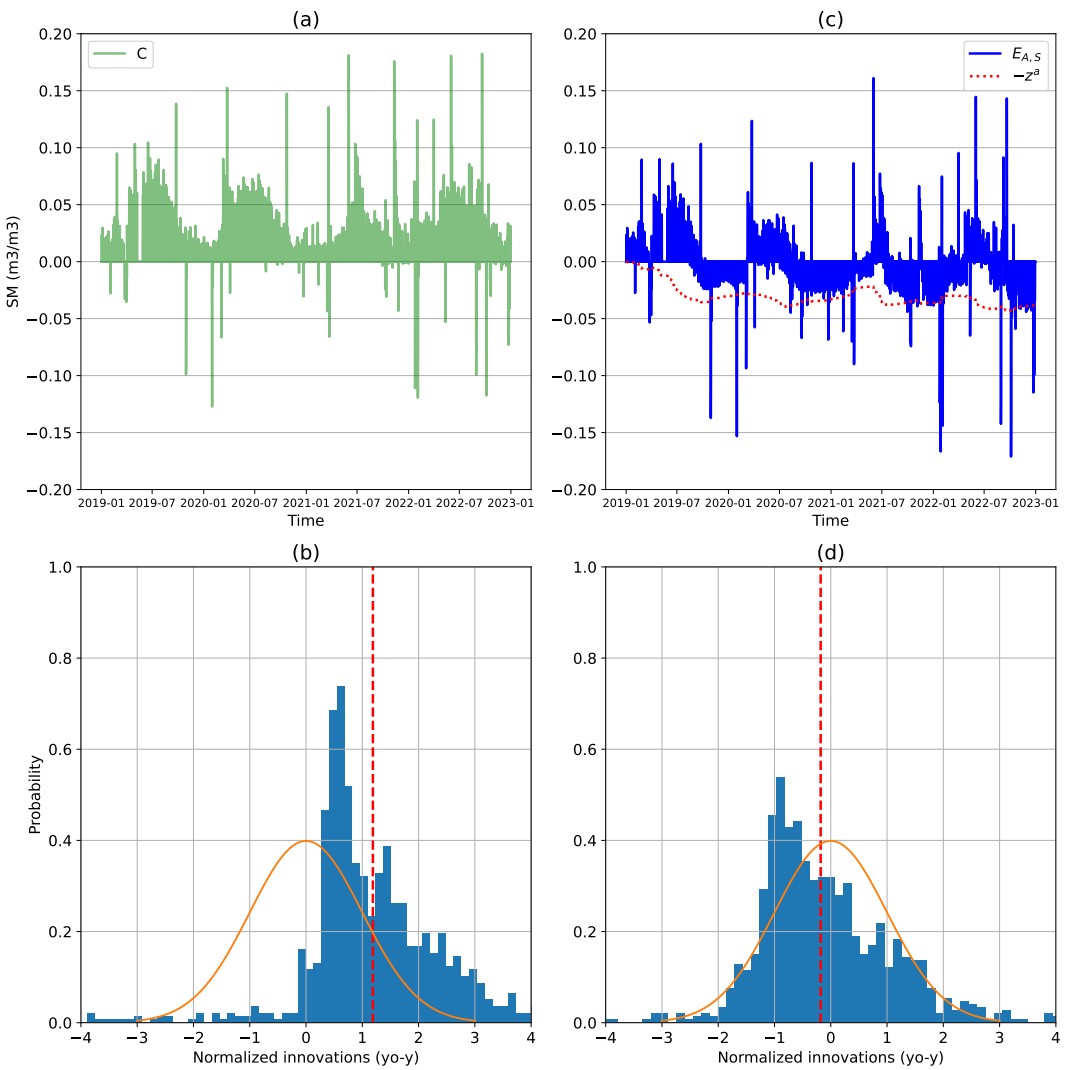

**Figure 5.** (**a**) Monthly mean SMOS SM departures ($m^3/m^3$) for C, corresponding to a point in Australia with latitude 25° S and longitude 140° E during 2019–2022. (**b**) Normalised innovations for C. The mean of the normalised departures is given by the vertical red dashed line. Plots (**c,d**) are equivalent to plots (**a,b**) but for experiment $E_{A,S}$.

Table 4 summarises the global average internal diagnostics for the five experiments. The results suggest that the ASCAT and SMOS biases were mainly independent of each other, as the BC of SMOS alone ($E_S$) did little to reduce the ASCAT SM departures and vice versa when assimilating the ASCAT alone ($E_A$). Experiment $E_{A,S}{}^*$ had larger absolute ASCAT SM departures than C, which suggests that the ASCAT adaptive BC and ASCAT seasonal rescaling complemented each other. The mean SSM increments were reduced by the combination of adaptive BC and ASCAT seasonal rescaling. Furthermore, the absolute values were slightly reduced. However, the impact of the adaptive BC on the RZSM increments was relatively small compared with the SSM increments. The 95% CIs confirm that the differences between the experiments are statistically significant.

**Table 4.** Global mean SM ($O − F$) departures (depar.) and SM increments (inc.) for the experiments ($m^3/m^3$). The 95% CIs are also included.

| Variable | C | $E_A$ | $E_S$ | $E_{A,S}$ | $E_{A,S}{}^*$ |
|---|---|---|---|---|---|
| ASCAT SM depar. ($\times 10^{-3}$) | $-5 \pm 0.1$ | $-1 \pm 0.1$ | $-5 \pm 0.1$ | $-1 \pm 0.1$ | $1 \pm 0.1$ |
| Absolute ASCAT SM depar. ($\times 10^{-3}$) | $26 \pm 0.03$ | $23 \pm 0.04$ | $26 \pm 0.03$ | $23 \pm 0.03$ | $29 \pm 0.04$ |
| SMOS SM depar. ($\times 10^{-3}$) | $-2 \pm 0.1$ | $-2 \pm 0.1$ | $3 \pm 0.1$ | $3 \pm 0.1$ | $3 \pm 0.1$ |
| Absolute SMOS SM depar. ($\times 10^{-3}$) | $37 \pm 0.03$ | $37 \pm 0.04$ | $32 \pm 0.03$ | $32 \pm 0.04$ | $32 \pm 0.03$ |
| SSM inc. ($\times 10^{-5}$) | $16 \pm 0.4$ | $13 \pm 0.4$ | $14 \pm 0.4$ | $10 \pm 0.3$ | $12 \pm 0.3$ |
| Absolute SSM inc. ($\times 10^{-5}$) | $204 \pm 0.3$ | $202 \pm 0.3$ | $201 \pm 0.3$ | $200 \pm 0.3$ | $202 \pm 0.3$ |
| RZSM inc. ($\times 10^{-5}$) | $4 \pm 0.2$ | $4 \pm 0.2$ | $3 \pm 0.2$ | $3 \pm 0.2$ | $3 \pm 0.2$ |
| Absolute RZSM inc. ($\times 10^{-5}$) | $98 \pm 0.3$ | $98 \pm 0.3$ | $97 \pm 0.3$ | $97 \pm 0.3$ | $97 \pm 0.3$ |

### 3.2. SM and ST Validation

The analysed SM and ST were validated using observations from the ISMN during 2020–2022. Figure 6 shows the locations of the stations for each network over the US, Australia, France, Spain and Germany. Also shown is the relative performance of the SSM Pearson R anomaly for $E_{A,S}$ compared with C. Overall, 1.7% of all the stations in $E_{A,S}$ significantly improved (at the 95% confidence level) on C, and 0.5% of the stations were significantly degraded relative to C, with all these sites located in the US (see Figure 6a). Table 5 gives the global average SSM and RZSM scores for the different experiments. There was a slight improvement in the SSM and RZSM anomaly correlations for the adaptive BC experiments compared with the control. The other scores were largely equivalent when averaged over all the networks. Whilst the impact of the BC was relatively small for most stations, there were some strong local impacts. For example, Figure 7 shows the time series of the SSM analysis for the different experiments at a point located in the USCRN network in the US, which corresponds to the station circled in Figure 6. In this case, the adaptive BC reduced the positive bias in the experiments and subsequently improved the fit to the observations, especially when both the ASCAT and SMOS were bias-corrected ($E_{A,S}$).

### 3.3. Atmospheric Validation

Figure 8 presents the global difference in the normalised relative humidity RMSE (dRMSE) for $E_S$ and $E_A$ with respect to C at 1000 hPa and for forecast lead times of 1–10 days. The impact of the adaptive BC, although small in both cases, varied across different regions, with the SMOS adaptive BC showing a slightly positive impact over the southern extratropics and the tropics and the ASCAT adaptive BC having a slightly negative impact over the extratropics of the Northern Hemisphere. Figure 9 shows a latitude–pressure plot of the difference in mean errors (dME) for $E_S$ (top), $E_{A,S}$ (middle) and $E_{A,S}{}^*$ (bottom) for a lead time of 72 h. The mean errors for $E_S$ were significantly improved in the lower troposphere (above 850 hPa) between 60 and 70 degrees north. The pattern was similar for $E_{A,S}$, but the signal was spread over a larger latitude band between 30 and 70 degrees north. On the other hand, the signal was reduced, going from $E_{A,S}$ to $E_{A,S}{}^*$. Together, these results demonstrate the complementary impacts of the adaptive BC and the ASCAT seasonal rescaling on the mean relative humidity in the boundary layer for the

midlatitudes of the Northern Hemisphere. Further work found the dME improvements persisted to about day 5. Figure 10 shows the mean differences in RH between $E_{A,S}$ and C. There were reductions in the relative humidity over eastern China and South America, which roughly correspond with the BC seen in Figure 3c,f, respectively. Likewise, the positive difference over the southeastern US agreed with the positive BC in Figure 3f. In the case of atmospheric temperature, neither the dME nor the dRMSE were significantly impacted by the adaptive BC.

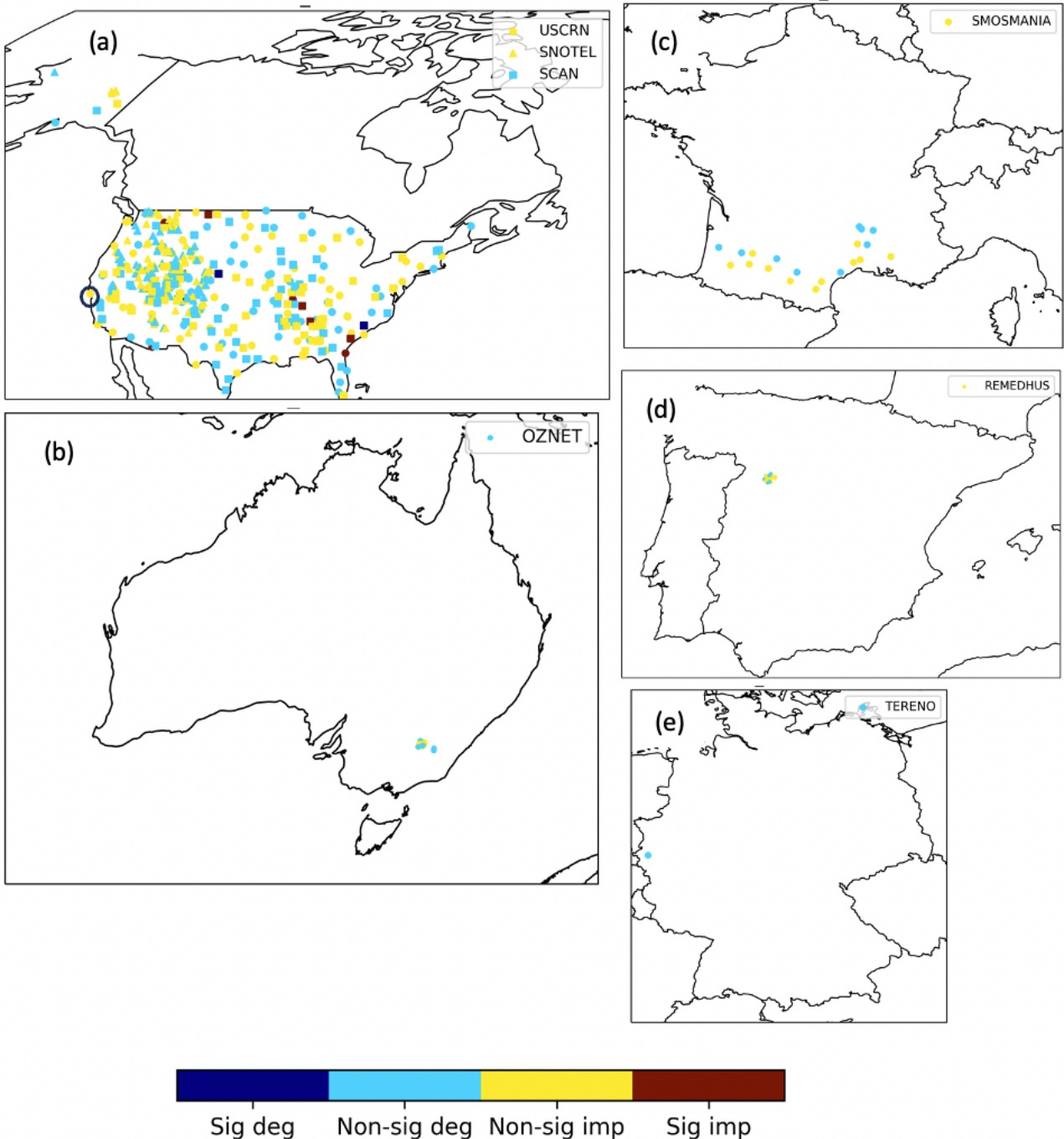

**Figure 6.** Locations of the ISMN networks used in the validation for (**a**) the US, (**b**) Australia, (**c**) France, (**d**) Spain and (**e**) Germany. Also shown is the relative Pearson R anomaly performance of $E_{A,S}$ compared with C for each station during 2020–2022, which correspond to significant (Sig) or non-significant (Non-sig) improvements (imp) or degradations (deg). The circled station in the US was used for the time series in Figure 7.

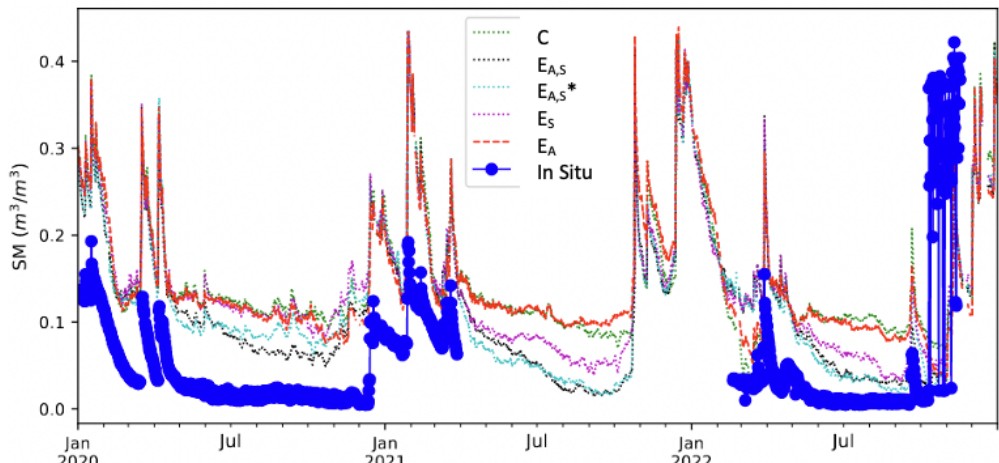

**Figure 7.** Time series of the SSM analysis (0–7 cm depth) for all the experiments over a point in the USCRN network of the US with latitude = 37.2° and longitude = −120.9°. Also shown are the in situ SM observations.

**Table 5.** Mean scores for surface soil moisture (SSM), root zone soil moisture (RZSM), surface soil temperature (SST) and root zone soil temperature (RZST) for the different experiments. The best scores are shown in bold font. The dimensionless units are expressed as (-).

| Global Mean Score | C | $E_A$ | $E_S$ | $E_{A,S}$ | $E_{A,S}{}^*$ |
|---|---|---|---|---|---|
| SSM R anomaly (-) | 0.439 | 0.439 | **0.441** | **0.441** | **0.441** |
| SSM UbRMSE ($m^3/m^3$) | 0.063 | 0.063 | 0.063 | 0.063 | 0.063 |
| SSM bias ($m^3/m^3$) | 0.065 | 0.065 | 0.065 | 0.065 | 0.065 |
| RZSM R anomaly (-) | 0.440 | 0.441 | 0.442 | **0.444** | 0.438 |
| RZSM UbRMSE ($m^3/m^3$) | 0.045 | 0.045 | 0.045 | 0.045 | 0.045 |
| RZSM bias ($m^3/m^3$) | 0.06 | 0.06 | 0.06 | 0.06 | 0.06 |
| SST R anomaly (-) | 0.675 | 0.675 | 0.675 | 0.675 | 0.675 |
| SST UbRMSE (°K) | 4.13 | **4.12** | 4.13 | **4.12** | 4.13 |
| SST bias (°K) | 0.64 | 0.64 | 0.64 | 0.64 | 0.64 |
| RZST R anomaly (-) | 0.630 | 0.630 | 0.630 | 0.630 | 0.630 |
| RZST UbRMSE (°K) | **2.25** | 2.26 | 2.26 | 2.26 | **2.25** |
| RZST bias (°K) | 0.74 | 0.74 | 0.74 | 0.74 | 0.74 |

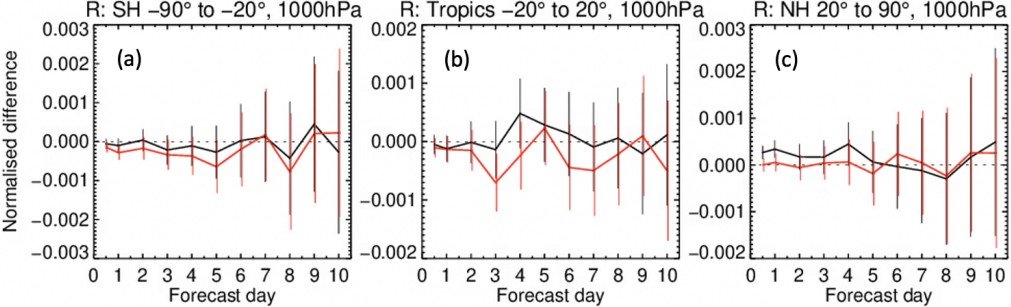

**Figure 8.** Normalised RMSE differences (dRMSE) for relative humidity at 1000 hPa over forecast days 1–10 for the experiments $E_A$ (black line) and $E_S$ (red line) compared with C, computed over 2020–2022. Results are shown for (**a**) Southern Hemisphere extratropics, (**b**) the tropics and (**c**) Northern Hemisphere extra-tropics.

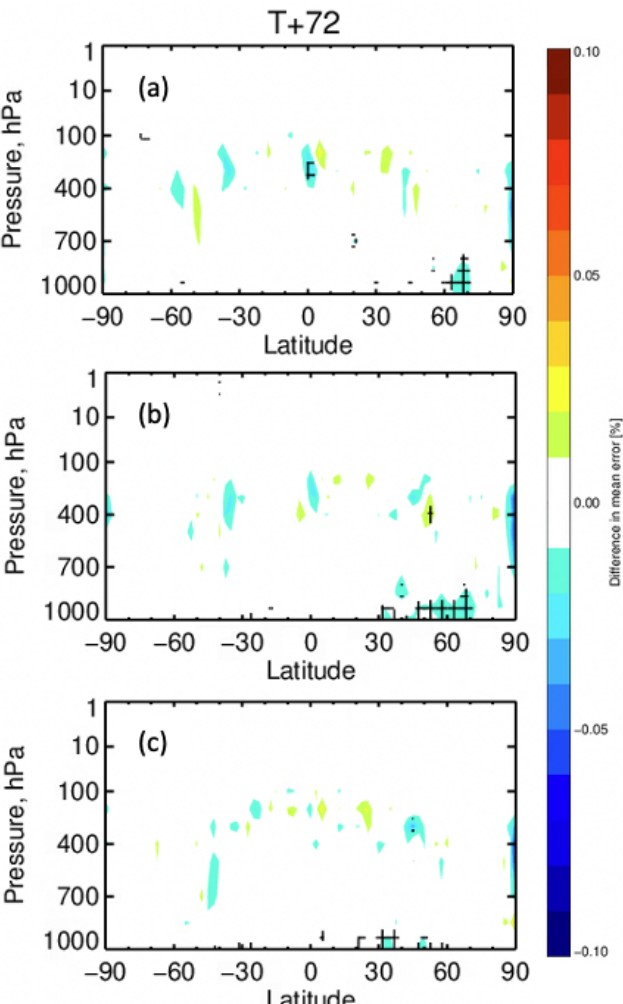

**Figure 9.** Latitude–pressure diagram of the mean forecast error differences (dME) in relative humidity (%) between the experiment and the control, averaged over 2020–2022: (**a**) experiment $E_S$, (**b**) $E_{A,S}$ and (**c**) $E_{A,S}$*. The forecast lead time was t + 72 h. Cross-hatching indicates statistical significance at the 95% confidence level.

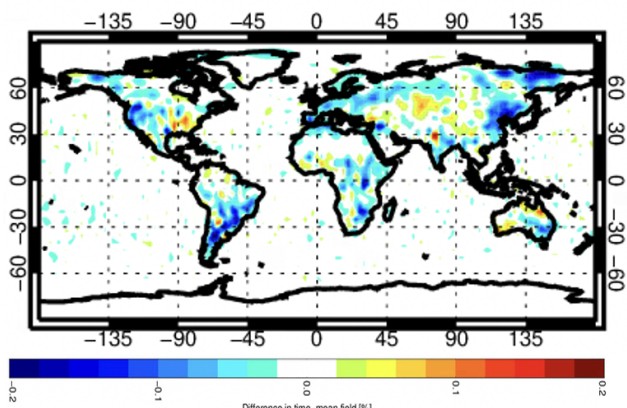

**Figure 10.** Difference in mean relative humidity (%) between $E_{A,S}$ and C at 1000 hPa for a forecast lead time of t + 48 h, averaged over 2020–2022.

## 4. Discussion

### 4.1. Adaptive Bias Correction vs. Seasonal Rescaling

The current seasonal rescaling for ASCAT SM accounts for stationary biases in the mean and variance. On the other hand, the adaptive BC accounts for non-stationary biases, but it only corrects the mean SM state. Therefore, the adaptive BC alone could not outperform the seasonal rescaling approach. However, when the adaptive BC was employed on top of the seasonal rescaling, the performance was improved substantially.

Global maps of the adaptive BC were plotted over August 2022 for the ASCAT and SMOS SM in Figure 3. For SMOS assimilation, the adaptive BC appeared to be more active in some regions than others. For example, over South Africa, the positive $(O - F)$ departures were reduced by the adaptive BC. On the other hand, over the Sahel and India, the negative departures were largely unchanged by the adaptive BC. Further evaluation determined that the negative summer biases over the Sahel and India had a shorter time scale than the effective memory of the adaptive BC. For the ASCAT SM, seasonal biases are removed by the seasonal rescaling, which is applied prior to assimilation. The SMOS NN training does not account for seasonal variability. These results indicate that it could be beneficial to include seasonal predictors in the SMOS NN.

### 4.2. Geographical and Temporal Variations in the Bias Correction

Over the four-year experiment period, a slight positive global trend was found for the ASCAT SM departures. Further analysis demonstrated that the positive trend could be largely attributed to moistening ASCAT observations. Whilst 4 years is a relatively short timescale, these results appear to be consistent with Hahn et al. [60], who demonstrated that the backscatter signal from ASCAT SM observations has been increasing in some regions over the last 15 years. This is partly related to land use change, including urbanisation and deforestation. They found that a periodic recalibration of the dry and wet backscatter reference can help to mitigate the spurious trend, and future ASCAT SM products will be recalibrated accordingly. In this study, the adaptive SM BC effectively removed a global positive trend in the ASCAT SM.

In the case of SMOS, large local SM biases in the $(O - F)$ departures were found over several regions, including a strong positive bias over the Andes in South America and over parts of Australia and a negative bias over the eastern US and high-latitude regions in Asia. The SMOS NN is trained globally, which may partly explain these results. The biases could be alleviated by introducing local predictors in the training dataset, such as latitude and longitude, which will be investigated in future versions of the NN. Nevertheless, the adaptive BC corrected many of these local biases, which was illustrated for a persistent positive bias over a point in eastern Australia.

In Figure 4, the average increments were plotted for August 2022. In general, the adaptive BC had the greatest impact on the increments in high-latitude regions, with relatively small impacts on lower latitudes. This is partly related to large local biases in high latitudes, especially in northeast Asia. These regions are known to have quality control issues related to frozen conditions, which could explain the larger biases. Further analysis has also found that high latitudes often coincide with areas where ascending and descending orbits overlap within the 12 h window. Thus, in the summer in the Northern Hemisphere, these regions often have two ASCAT and SMOS observations per grid point in the 12 h assimilation window. Future work should investigate the impact of the observation density on the analysis increments and the adaptive BC.

### 4.3. Performance Validation

The SM and ST analyses were validated using in situ data from over 500 stations, representing a range of different climates and vegetation types over the US, Europe and Australia. Statistically significant impacts were only found for 10 sites located in the US. For the SMOS $(O - F)$ departures, the US was affected by a strong positive (negative) bias on the western (eastern) side during August 2022 (see Figure 3). Thus, the adaptive BC

corrected these biases and modified the SM analysis over these regions. However, many regions could not be validated due to the lack of in situ data, including northeast Asia, where some of the largest biases were corrected. Further work should also determine whether the climate and the land cover type influence the effectiveness of the adaptive BC.

The 1–10 day NWP forecasts of the atmospheric temperature and relative humidity were validated against the ECMWF operational analysis. It is well known that the SM analysis has an important influence on latent and sensible heat fluxes between the surface and the boundary layer (see, for example, Fairbairn et al. [35]). Interestingly, in this study, the adaptive BC improved the mean relative humidity forecasts but not the temperature forecasts. This could be related to the atmospheric conditions or model deficiencies. A tool is being developed at the ECMWF to validate the latent and sensible heat fluxes directly using FLUXNET data [61], which should help to understand these results.

### 4.4. Assumptions in the Bias Correction Approach

Whilst it is acknowledged that the perfect model assumption in this study is likely to be incorrect, the heterogeneity of SM together with the lack of an accurate ground reference makes it extremely challenging to differentiate between model and observation errors in global land data assimilation (DA) systems. Furthermore, land surface models rely on calibrated characterisations to relate SM prognostic variables to model fluxes, such as hydraulic conductivity and latent or sensible heat exchanges with the boundary layer. Arguably, accounting for modelled SM biases can only be achieved by correcting the model parameters in conjunction with the state estimates [13,62]. Joint initiatives between the ECMWF and other organisations are working to improve land surface benchmarking, such as the PLUMBER2 model inter-comparison project. In the work of Abramowitz et al. [63], various different local land surface model simulations were evaluated using data from eddy covariance-based flux tower sites around the world. Furthermore, empirical models were used as a benchmark to understand mechanistic model uncertainties.

The formulation of the two-stage filter in this study assumes that the ASCAT and SMOS biases are uncorrelated. Accounting for bias correlations could be achieved with state augmentation, but this would significantly increase the complexity and computational cost of the filter. With the exception of northeast Asia, the spatial patterns of the biases in Figure 3 for August 2022 are generally different between the two observation types, which supports this assumption. For simplicity, the study also assumes that the T2m and RH2m observations are bias-free. Future works could investigate a similar adaptive BC approach for T2m and RH2m observations, which could complement the adaptive BC of the SM observations.

## 5. Conclusions

In this study, a novel adaptive SM BC approach was evaluated in the ECMWF land DA system over the period of 2020–2022. A two-stage filter was formulated to bias-correct both the ASCAT and SMOS level 2 SM observations. Compared with the existing SEKF SM analysis, the two-stage filter was computationally efficient and only required the additional storage of the ASCAT and SMOS biases for each grid point. Furthermore, the adaptive BC was designed to complement the operational SMOS NN and the ASCAT seasonal rescaling BC approaches.

Experiments were set up to assess the impact of the two-stage filter on ASCAT and SMOS SM assimilation. Firstly, the internal DA diagnostics were evaluated in terms of the mean (observation−forecast) departures and the SM increments for the different experiments. The magnitude of the departures was reduced by up to 20% for the ASCAT SM. The greatest impact occurred when the ASCAT observations were most active during the summer in the Northern Hemisphere. Furthermore, up to 6% more ASCAT observations were assimilated in the adaptive BC experiment as excessive SM departures were removed less often by the quality control. For the SMOS SM, the adaptive BC reduced the magnitude

of the global departures by up to 30%, with typically 10% more observations assimilated in the Northern Hemisphere during the summer months.

The impact of the adaptive BC on SM and ST analysis performance was evaluated using observations from the international soil moisture network. On average, the adaptive BC slightly improved the anomaly correlation of the surface and root zone SM with the observations. There were also some locally significant impacts for individual stations. Nevertheless, the adaptive BC experiments demonstrated statistically significant improvements over the control in the 1–5 day forecasts of the mean relative humidity in the lower boundary layer, mainly over the mid-to-high latitudes in the Northern Hemisphere.

Finally, it is planned that the adaptive BC will be implemented operationally at the ECMWF. A simplified version of the IFS was adopted for the experiments in this study in order to minimise the computational cost. A further evaluation of the adaptive SM BC will be carried out using the full weakly coupled land–atmosphere DA system.

**Author Contributions:** Conceptualisation, methodology, software, validation, formal analysis, investigation, resources, data curation, writing—original draft preparation and writing—review and editing, D.F., P.d.R. and P.W.; supervision, P.d.R. All authors have read and agreed to the published version of the manuscript.

**Funding:** D.F. was funded by the EUMETSAT Satellite Application Facility on Support to Operational Hydrology and Water Management (H SAF CDOP4, contract EFP0291). P.W. was funded by the European Space Agency (ESA) SMOS Expert Support Laboratories (ESL), and P.d.R. received ECMWF core funding.

**Data Availability Statement:** The data presented in this study are available on request from the corresponding author. The data are not publicly available due to restrictions on public access to research experiments in the ECMWF archiving facility.

**Acknowledgments:** The authors thank the three anonymous reviewers and the editor for their constructive comments, which greatly improved the quality of the manuscript.

**Conflicts of Interest:** The authors declare no conflicts of interest.

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
