# Peer review of "Evaluation of an Adaptive Soil Moisture Bias Correction Approach in the ECMWF Land Data Assimilation System"

_remotesensing, doi:10.3390/rs16030493_

Round 1
Reviewer 1 Report
Comments and Suggestions for Authors
I have reviewed the manuscript remotesensing-2778203 entitled “Evaluation of an adaptive soil moisture bias correction approach in the ECMWF land data assimilation system”. In this study, authors evaluate the two-stage filter to dynamically correct soil moisture biases from satellite-derived active ASCAT C-band and passive L-band SMOS surface soil moisture observations in the European Centre for Medium-Range Weather Forecasts (ECMWF) land data assimilation system. The authors have done a very comfortable job in providing the explanation and the results. This paper deserves to be published but there are some minor issues that should be clarified before the paper can be accepted.
1. From the abstract alone, it is hard to understand what was done in this study and with what aim. It should be re-written. Overall, the motivation of this work needs to come out more clearly.
2. The manuscript should also be improved in terms of the application of punctuation marks.
3. The novelty of the paper is completely missing. The reason behind the use of ECland surface model is completely missing.
4. The narrative used throughout the introduction section and the flow of the content is not appropriate.
5. The novelty/originality of the paper should be more effectively established. It would be advisable to add a Table to the “Introduction” section, tabulating the latest research works in the field to highlight the novelty of the present work accordingly. This should also help with shortening this section by moving some of the content to this table.
6. It’s essential to include equation numbers and related references for the equations mentioned in the manuscript. Equation numbers provide clarity and ease of reference for readers and reviewers, helping to ensure a better understanding of the content.
7. The limitations of the study should also be explained under a new section as “Limitations of the present study” right before the summary and discussion.
Comments on the Quality of English LanguageMinor editing of English language required.
Author Response
Please see file attached

Reviewer 2 Report
Comments and Suggestions for Authors
The research work under consideration is quite compelling, showcasing a methodologically sound approach and a meticulous scientific analysis of the data. However, there are a few minor modifications that warrant attention and should be addressed for further refinement and clarity in the study.
· The literature review section should be more comprehensive, delving into the existing research with greater depth. Additionally, it is crucial to explicitly highlight the identified research gap within the introduction section.
· In the introduction section, it is essential to include clear and concise objectives to guide the reader.
· Please remove lines 56-63 as they are not necessary for the content.
· Consider incorporating a flowchart of the methodology to enhance the overall understanding of the research approach.
· Dedicate a separate section to the discussion, emphasizing critical analysis and further elaborating on the validation status from various perspectives.
· Lastly, instead of a summary, include a conclusive section where a robust conclusion is drawn, encapsulating the key findings of the study and their implications.
Comments on the Quality of English LanguageMinor editing of English language required
Author Response
Please see file attached

Reviewer 3 Report
Comments and Suggestions for Authors
Dear Authors, kindly check the English sentence structure and grammar errors throughout the manuscript, also suggest to modified changes as per the technical comments given below to enhance the clarity coherence, and overall quality of the manuscript.
· Line 3; The title is clear and concise, providing a good overview of the study's main focus and methodology. It appropriately highlights the key elements: the adaptive soil moisture bias correction approach, its evaluation, and the context of the ECMWF land data assimilation system.
· Line 1; Consider simplifying technical terms for a wider audience.
· Line 2-3; which machine learning approach mentioned some technics name in the abstract.
· Line 7-9; It would be helpful to include a sentence on the methodology used for the evaluation (around to give the reader a better understanding of how the results were obtained.
· Line 10-11; The mention of specific satellite systems (ASCAT and SMOS) and the percentage improvement in first guess-observation departures adds valuable specificity.
· Line 13-14; author should perhaps briefly mention the key challenge or problem statement for context.
· Line 27-29; it might be beneficial to briefly discuss how the mentioned approaches are specifically adapted or differ in the context of land surface models.
· Line 32-38; the traditional methods (CDF matching, machine learning) and their limitations is well-placed, but should provide a clear rationale for the need for an adaptive approach.
· Line 48-50; The background on ECMWF’s current practices with ASCAT and SMOS provides valuable context and as suggestion It would be helpful to briefly discuss how these practices compare with those at other major weather forecasting centers.
· Line 48-54; author should brief explanation of why SM bias-correction approach is expected to be more effective than existing approaches would strengthen the rationale for this study.
· Line 37: Suggest rephrasing for better readability: "Whilst these methods work well in many practical applications, they require large calibration/training datasets, and the assumption of stationary biases is often suboptimal." Breaking the sentence into two parts enhances clarity.
· Line 56-60, should rewrite the statement and write a specific objective with addressing challenge of the study.
· Line 17-63, overall suggestion to author for adding some more details in introduction section for improvement of manuscript for keeping the standard of journal.
· Line 67: Suggest rephrasing for clarity: "The operational SM analysis is based on a point-wise SEKF with 12-hour assimilation windows, implemented in 2010 to assimilate proxy screen-level observations of 2m temperature (T2m) and relative humidity (RH2m)." This would enhance readability.
· Line 83: The simplification of the observation operator is noted Hi​(xb​), but it would be useful to discuss any potential limitations or assumptions inherent in this simplification, especially regarding the representation of soil moisture at different depths.
· Line 106-108: The assumption that proxy SLV observations are bias-free and the separate treatment of ASCAT and SMOS NN biases are critical points, while it would be beneficial to discuss the justification for these assumptions and how they might affect the results.
· Line 131-133: a technical discussion on the impact of this averaging approach on the accuracy and reliability of the bias correction would be insightful for the manuscript.
· Line 143-147: The description of ASCAT data processing is concise. However, a more detailed explanation of the change detection approach and its implications for soil moisture estimation would be valuable.
· Line 180: Recommend clarifying the sentence: "The NN is trained globally, which means that while the global biases between the SMOS SM observations and the model SM are small, significant regional biases remain."
· Line 186: Suggest rephrasing for clarity: "Each gridpoint is divided into 8 tiles, representing different land cover types, including vegetation types, soil, and snow cover, using data from the US Department of Agriculture Global Land Cover Climatology (GLCC) map." This enhances readability.
· Line 189: author should discuss how these 4 vertical layers of moisture interact and representation compares to more complex or simplified models would be informative.
· Line 200-202: The use of SSA for long-term experiments is mentioned, however, a technical evaluation of the advantages and limitations of using SSA compared to fully coupled DA approaches in the context of soil moisture analysis would be insightful.
· Line 215: Recommend rewording for clarity: "(i) C is the control with the operational configuration of ASCAT and SMOS assimilation, without adaptive BC." This enhances the description of the control experiment.
· Line 215-220: Authors should add more detailed rationale behind the specific configurations of each experiment, especially the choice of parameters and their expected impact on the results, would add depth to this section.
· Line 243-245: The challenges of using in situ data for validation are acknowledged, but should mention a technical discussion on how these challenges were addressed or mitigated in this study would be useful.
· Line 270: Recommend rewording for better readability: "The upper air verification is performed by comparing the forecasts against the ECMWF operational analysis." This simplifies the sentence structure.
· Line 280-288: The evaluation of the adaptive BC on ASCAT SM first guess departures is well presented in manuscript, but suggested to add, some more detailed analysis of the statistical significance of the 10-20% reduction in departures would strengthen the findings. Additionally, insights into the potential reasons for the year-long spin-up period for bias-correction would be valuable.
· Line 289-290: Recommend rewording for clarity: "Up to 6% more observations are assimilated in EA,S compared to C, with the largest difference observed during the northern hemisphere summer/autumn, a period of active ASCAT assimilation in this region." This ensures better understanding of the observation count differences.
· Line 308-319: author should provide a deeper analysis of why certain regions like India and the Sahel are less responsive to the adaptive BC would add value, possibly exploring the role of local climatic or land surface characteristics.
· Line 320-329: suggested to added technical discussion on the potential reasons for the smaller magnitude increments in these regions and their implications for soil moisture modeling would be beneficial.
· Line 359-364: The mention of strong local impacts of the BC is interesting. It would be useful to have a more detailed analysis of these local impacts, possibly including a discussion on the variability in the effectiveness of the BC across different geographical locations.
· Line 367-368: Consider rephrasing for better readability: "The impact of the adaptive BC, although small in both cases, varied across different regions, with the SMOS adaptive BC showing a slightly positive impact over the southern extratropics and the tropics, and the ASCAT adaptive BC having a slightly negative impact over the northern hemisphere extratropics."
· Line 379-384: The lack of significant impact on atmospheric temperature by the adaptive BC is an interesting finding in your manuscript, but mentioned the reason on why the BC might have a more pronounced effect on humidity than temperature, considering the land-atmosphere interactions, would add depth to the analysis.
· Line 392-393: Suggest rephrasing for precision: "The impact of the two-stage bias filter on the internal DA diagnostics was assessed for ASCAT and SMOS SM by evaluating the first guess (observation-model) departures and the SM increments.
· Line 433-448: The discussion on the challenges of differentiating between model and observation errors in global land DA systems is important. Further elaboration on the potential approaches to address these challenges, such as joint initiatives for improving benchmarking datasets, would provide a comprehensive view of the future directions in this field.
· Line 385: Author should split the section into Discussion and Conclusions/summary.
· Author should ensure that all references, if any, are correctly formatted and consistent throughout the manuscript according to journal style or follow the instructions on https://www.mdpi.com/journal/remotesensing/instructions.
Comments on the Quality of English Language· Line 37: Suggest rephrasing for better readability: "Whilst these methods work well in many practical applications, they require large calibration/training datasets, and the assumption of stationary biases is often suboptimal." Breaking the sentence into two parts enhances clarity.
·Line 67: Suggest rephrasing for clarity: "The operational SM analysis is based on a point-wise SEKF with 12-hour assimilation windows, implemented in 2010 to assimilate proxy screen-level observations of 2m temperature (T2m) and relative humidity (RH2m)." This would enhance readability.
·Line 186: Suggest rephrasing for clarity: "Each gridpoint is divided into 8 tiles, representing different land cover types, including vegetation types, soil, and snow cover, using data from the US Department of Agriculture Global Land Cover Climatology (GLCC) map." This enhances readability.
·Line 367-368: Consider rephrasing for better readability: "The impact of the adaptive BC, although small in both cases, varied across different regions, with the SMOS adaptive BC showing a slightly positive impact over the southern extratropics and the tropics, and the ASCAT adaptive BC having a slightly negative impact over the northern hemisphere extratropics."
·Line 392-393: Suggest rephrasing for precision: "The impact of the two-stage bias filter on the internal DA diagnostics was assessed for ASCAT and SMOS SM by evaluating the first guess (observation-model) departures and the SM increments.
